# Hypoosmotic stress induces flagellar biosynthesis and swimming motility in *Escherichia albertii*

Tetsuya Ikeda[1✉], Toshie Shinagawa[2], Takuya Ito[1], Yuta Ohno[1], Akiko Kubo[1], Junichiro Nishi [3], Yasuhiro Gotoh[4], Yoshitoshi Ogura[4], Tadasuke Ooka [3] & Tetsuya Hayashi[4]

Bacteria use flagella as propellers to move to favorable environments. *Escherichia albertii*, a growing cause of foodborne illness and diarrhea, is reportedly non-motile and lacks flagella on its surface. Here, we report that 27 out of 59 *E. albertii* strains, collected mainly from humans and birds, showed swimming motility when cultured at low osmotic pressure. The biosynthesis of flagella in *E. albertii* cells was induced under ambient temperature and hypoosmotic pressure: conditions which resemble aquatic environments. Flagellar induction increased *E. albertii* survival in the intestinal epithelial cell culture containing gentamicin. Although genes involved in chemotaxis are not present in the *E. albertii* genome, the addition of glutamic acid, an amino acid known to regulate the internal cell osmolarity, augmented the proportion of swimming cells by 35-fold. These results suggest that flagellar biosynthesis and motility in *E. albertii* cells are controlled by their internal and external osmolarity.

[1] Hokkaido Institute of Public Health, Kita-19, Nishi-12, Kita-ku, Sapporo 060-0819, Japan. [2] Regenerative Medicine Laboratory, Nozaki Tokushukai Hospital Research Institute, 2-10-50 Tanigawa, Daito 574-0074, Japan. [3] Department of Microbiology, Graduate School of Medical and Dental Sciences, Kagoshima University, 8-35-1 Sakuragaoka, Kagoshima 890-8544, Japan. [4] Department of Bacteriology, Faculty of Medical Sciences, Kyushu University, 3-1-1 Maidashi, Higashi-ku, Fukuoka 812-8582, Japan. ✉email: ikeda@iph.pref.hokkaido.jp

*E*scherichia albertii is an emerging enteropathogen causing diarrhea and gastroenteritis in humans. *E. albertii* strains carry several virulence genes such as the *eae* gene, encoding intimin; two sets of genes encoding two different type III secretion systems (T3SSs); and the *cdtB* gene, encoding the cytolethal distending toxin[1–3]. In some strains, the *stx2a* and *stx2f* genes for Shiga toxins have also been found[4,5].

*E. albertii* was first isolated from diarrheal stools of children in Bangladesh[1]. Since then, *E. albertii* has been isolated from patients of many foodborne illness outbreaks, including that which occurred in Okinawa, Japan, in 2016, where 217 persons suffered from diarrhea and/or abdominal pain after consumption of a salad[6]. *E. albertii* has also been isolated from wild birds[2], felines[5], swine[7], and chickens[8]. Notably, *E. albertii* is frequently isolated from birds, suggesting its ability to colonize them.

One of the habitats of *E. albertii* is assumed to be the gastrointestinal tract of birds and mammals. Once excreted from the gut, *E. albertii* cells face a hostile external environment. Most pathogenic bacteria infecting the gastrointestinal tract are motile[9]. In liquid medium, they use flagella for swimming at speeds of 10–200 $\mu m\,s^{-1}$. Upon nutrient depletion, γ-proteobacteria including *Plesiomonas shigelloides* and *Vibrio cholerae* eject their flagella at the base of the flagellar hook to halt costly motility[10]. *Vibrio alginolyticus* downregulate flagella-related genes in response to heavy metal stress or low pH[11]. Understanding how *E. albertii* cells sense and circumvent such harsh conditions is critical to understanding bacterial survival strategies and developing methods for preventing their contamination and proliferation in food items.

The directed, active movement towards or away from specific chemical substances is called chemotaxis. Motility and chemotaxis are well-recognized virulence factors in many pathogens[12,13]. Bacteria, such as *E. coli*, contain a full suite of genes for flagellar biosynthesis and chemotaxis[9]; however, *E. albertii* has been described as non-motile and non-flagellated, and lacks the genes for chemotaxis, despite the fact that 74% of *E. albertii* strains possess a complete set of flagellar biosynthesis genes[3]. Since flagella are energetically expensive to synthesize and operate, we postulated that *E. albertii* produces flagella under certain conditions.

In this study, we demonstrate that *E. albertii* cells produce flagella and exhibit swimming ability when they are grown at 20 °C and under low osmotic pressure in the culture medium. We also report that the addition of glutamic acid enhances the motility of *E. albertii*.

## Results

**Flagella formation and motility in *E. albertii*.** Since most *E. albertii* strains carry flagella-related genes and express their mRNAs[3], we investigated various culture conditions to find out if any of them may induce their motility. First, we tested a reduced set of *E. albertii* strains ($n = 6$). When these strains were cultured in semi-solid tryptic soy agar at 20 °C, no migration was observed. However, when cultured in the semi-solid agar containing diluted tryptic soy broth (TSB) (1/10), four strains formed an umbrella-like lateral extension at the upper part of the stab lines, revealing bacterial growth (Fig. 1a). This growth pattern was not observed when these strains were grown at 37 °C in the same medium. These results indicate that exposure to low nutrient and salt concentrations under relatively low temperature (20 °C) and aerobic conditions induced cell motility, at least in these four *E. albertii* strains.

The observation of these four strains using differential interference contrast microscopy confirmed that a substantial proportion of HIPH08472 cells swam in the diluted TSB

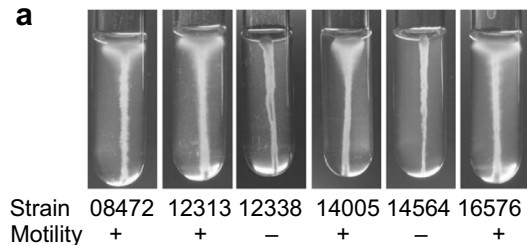

a

| Strain | 08472 | 12313 | 12338 | 14005 | 14564 | 16576 |
|--------|-------|-------|-------|-------|-------|-------|
| Motility | + | + | − | + | − | + |

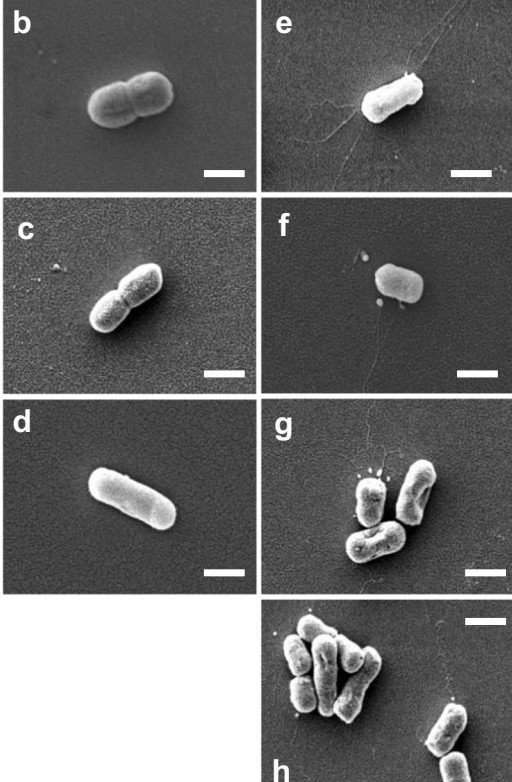

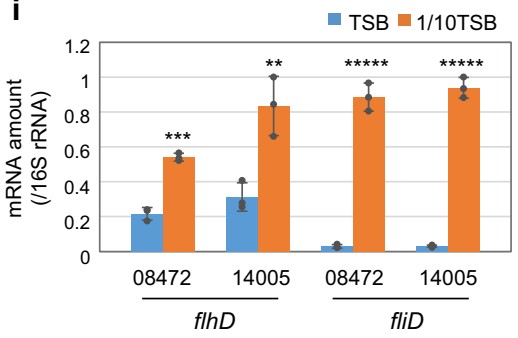

i

(1/10TSB) (Table 1). They exhibited a run-and-tumble motion similar to that of *E. coli*, wherein almost straight swimming segments were randomly interrupted by a sudden change in direction. Similar results were obtained for HIPH12313, HIPH14005, and HIPH16576 (Table 1, Supplementary Movie 1). Next, we examined the presence of flagella on the cells using a scanning electron microscope. The four motile strains had peritrichous flagella, whereas the two non-motile strains did not (Fig. 1b–h). To test whether flagella production could be affected by culture conditions, HIPH08472 cells grown in ordinary TSB or diluted TSB were examined to detect the presence of flagella. The cells cultured in ordinary TSB did not show flagella (Fig. 1b),

**Fig. 1 Induction of flagella production and motility in *E. albertii* by low salt and low nutrient concentrations. a** Cells of *E. albertii* strains were cultured in semi-solid 1/10TSB at 20 °C for 3 days. Four strains (HIPH08472, HIPH12313, HIPH14005, and HIPH16576) showed "umbrella-like" growth at the upper part of the stab lines, indicating that they are motile and aerobic condition enhances their motility. Two strains (HIPH12338 and HIPH14564) were non-motile and grew only along the stab lines. **b**–**h** Scanning electron microscopy images of *E. albertii* strains cultured in TSB (**b**) or 1/10TSB (**c**–**h**) at 20 °C. No flagella were observed in HIPH08472 cells cultured in TSB (**b**), while several flagella were clearly observed on both pole and lateral when cells were cultured in 1/10TSB (**e**). Flagella were also observed in other motile strains such as HIPH12313 (**f**), HIPH14005 (**g**), and HIPH16576 (**h**), but not in the non-motile strains HIPH12338 (**c**) and HIPH14564 (**d**). Scale bar: 1 μm. **i** Induction of the flagellar genes *flhD* and *fliD* by low-salt and low-nutrient levels. The amount of mRNA was measured by qRT-PCR. Data are shown as means ± s.d. ($n =$ 3 biologically independent samples). $**P < 0.01$, $***P < 0.001$, $*****P < 0.00001$.

while approximately 50% of the cells cultured in 1/10TSB presented several flagella on their pole and lateral surfaces (Fig. 1e).

**Non-motile strains show alterations in flagellar genes**. To further examine the transcriptional regulation of flagella production, mRNA levels of two flagella-related genes were analyzed by qRT-PCR in two motile strains. When cultured in 1/10TSB for 24 h, the expression of *fliD* and *flhD* (mRNA) was increased in both strains by about 30- and 2.5-fold, respectively (Fig. 1i). These results indicate that flagella production is regulated at the transcription level.

To clarify the genetic features of each strain, we sequenced the whole genome of these six strains (Table 1). In line with the results described, while the four motile strains contained a full set of non-altered flagellar biosynthesis genes (41 genes), the two non-motile strains contained multiple mutations in the flagellar biosynthesis genes. HIPH12338 had deletions in the flagellar hook-filament junction genes (*flgK* and *flgL*) and lacked the flagellar chaperone gene *filT*. HIPH14564 had 1-bp deletions in the flagellar MS ring component gene *filF* and the flagellar chaperone gene *filS*; transposon insertions in genes *flgN* and *flgM* were also detected. The non-motility of these two strains can be explained by the aforementioned mutations, given that flagellar assembly is hindered by defects in these genes.

**Flagellated *E. albertii* can circumvent gentamicin treatment**. To determine whether flagella affect virulence in *E. albertii*, the adhesion of flagella-induced and non-induced HIPH14005 and HIPH12338 cells to the human intestinal epithelial cell line Caco-2 and survival were evaluated in the presence of gentamicin

(Table 2). After 2 h of infection, the number of adherent bacterial cells on Caco-2 cells did not markedly different between the strains or the culture conditions tested. The number of surviving cells after gentamicin treatment in flagella-induced HIPH14005 cells was 130-fold that of non-induced cells. Surviving bacteria were rarely detected in Caco-2 cells infected with HIPH12338 cells, which have the same type of *eae* as HIPH14005 cells and mutations in flagellar genes. These results indicate that the flagella promote *E. albertii* invasion in intestinal epithelial cells, thereby increasing bacterial survival in the presence of gentamicin. However, bacterial resistance to gentamycin may increase upon culturing cells in 1/10TSB at 20 °C.

**Motility is a prevalent trait among *E. albertii* strains**. The other 53 *E. albertii* strains that we had collected were further tested for motility. They included 28 strains already sequenced in our previous study[3]. Of the 59 strains examined in total, 27 (45.8%) showed umbrella-like growth and swimming ability (Table 3). These data clearly indicate that motility is a prevalent trait in *E. albertii*. One strain showed motility only in the liquid medium. Another strain became motile after a longer incubation period (96 h). Out of the 28 previously sequenced strains, 5 contained some mutations in the flagellar biosynthesis genes and were non-motile. Most of the remaining sequenced strains (19/23) possessed an apparently intact gene set and were motile, but four strains were not. More detailed examinations are required to determine expression levels and patterns of flagellar biosynthesis genes in these four strains.

**Osmotic pressure regulates motility in *E. albertii* cells**. To determine whether specific components of TSB are responsible for motility induction in *E. albertii* cells, we prepared and tested various culture media in which each of the five TSB components (trypticase peptone, soybean meal, glucose, NaCl, and $K_2HPO_4$) was individually restricted. A strain displaying moderate motility (HIPH08472) and a second one displaying high motility (HIPH14005) were separately inoculated in these culture media. After 24 h of incubation, the proportions of swimming cells in the ordinary TSB were 0% and approximately 10% for HIPH08472 and HIPH14005, respectively. Surprisingly, in TSB containing 1/10 of the regular NaCl amount, the proportions of swimming cells increased to ≥18% for HIPH08472 and ≥28% for HIPH14005 (Fig. 2a). Besides, ≥1% of HIPH08472 cells began to swim in the culture restricted in $K_2HPO_4$. Based on these findings, we postulated that osmotic pressure rather than any particular substance regulates motility in *E. albertii* cells.

HIPH14005 cells were subsequently cultivated in media with increasing osmolarities (0.00–0.32 osm l$^{-1}$) achieved by the addition of NaCl, KCl, $MgSO_4$, or xylose to 1/10TSB (Fig. 2b). The proportion of swimming cells decreased from an average of 85% (at 0.00 osm l$^{-1}$) to less than 22% (at 0.32 osm l$^{-1}$),

**Table 1 Induction of motility in *E. albertii* by low salt and nutrient concentrations and low temperature.**

| Strain | Umbrella-like growth | | | Swimming | Flagella | Mutations in flagellar biogenesis genes |
|---|---|---|---|---|---|---|
| | Semi-solid TSB 20 °C | Semi-solid 1/10TSB 37 °C | Semi-solid 1/10TSB 20 °C | 1/10TSB 20 °C | 1/10TSB 20 °C | |
| HIPH08432 | − | − | + | + | + | − |
| HIPH12313 | − | − | + | + | + | − |
| HIPH12338 | − | − | − | − | − | *flgK*, *flgL*, *fliT* (deletion) |
| HIPH14005 | − | − | + | + | + | − |
| HIPH14564 | − | − | − | − | − | *fliS*, *fliF* (1 bp deletion) |
| HIPH16576 | − | − | + | + | + | − |

**Table 2 Bacterial adhesion on Caco-2 cells and survival after gentamicin treatment.**

| Strain | | HIPH14005 | | HIPH12338 | |
|---|---|---|---|---|---|
| | | 1/10TSB, 20 °C | TSB, 37 °C | 1/10TSB, 20 °C | TSB, 37 °C |
| Motility | | + | − | − | − |
| Number of associated bacteria (CFU) | exp1 | 2.1.E+06 | 1.1.E+06 | 1.1.E+06 | 4.3.E+05 |
| | exp2 | 2.8.E+06 | 5.7.E+05 | 1.0.E+06 | 1.0.E+06 |
| | exp3 | 2.4.E+06 | 1.2.E+06 | 1.5.E+06 | 1.2.E+06 |
| Number of surviving bacteria (CFU) | exp1 | 1.6.E+03 | <10 | <10 | <10 |
| | exp2 | 1.1.E+03 | 10 | <10 | 20 |
| | exp3 | 1.2.E+03 | <10 | <10 | 20 |

**Table 3 Motility of 59 E. albertii strains under flagella-inducing conditions.**

| Umbrella-like growth | Swimming | Number of strains | Proportion (%) |
|---|---|---|---|
| + | + | 27 | 45.8 |
| − | + | 1 | 1.7 |
| − | − | 31 | 52.5 |

irrespective of the substance used. These results indicate that osmotic stress represses motility in *E. albertii*.

To extend our analysis, we also prepared and tested chemically defined media on the basis of M9 minimal medium, which contains just salts and glucose. *E. albertii* cells of both HIPH08472 and HIPH14005 strains showed no growth or very slow proliferation in M9 minimal medium ($0.24 \, \text{osm} \, l^{-1}$) at 20 °C. However, the number of bacteria increased after 24 h of incubation by 20- to 30-fold when a mixture of essential amino acids was added (Fig. 3a), but the number of swimming cells was negligible. Thus, the M9 minimal medium supplemented with essential amino acids (M9E medium) was used as the basal medium for subsequent motility measurements.

Next, the effects of non-essential amino acids on motility were examined. When all of the seven non-essential amino acids were added to the M9E medium, the proportion of swimming cells increased from 0.33% to 20% (60-fold) in HIPH14005 cultures and from 0% to 10% in HIPH08472 cultures, indicating that non-essential amino acids are required to induce motility in *E. albertii*. To identify the amino acid responsible for motility induction, each of the seven non-essential amino acids was added separately to the M9E medium and then HIPH08472 or HIPH14005 cells were inoculated. We detected that the addition of glutamic acid increased the proportions of swimming cells to an average of 3% for HIPH08472 and 11% for HIPH14005. The addition of proline slightly increased the proportion of swimming cells in both strains (on average, 1% and 2%, respectively), whereas asparagine addition only affected strain HIPH14005. Although limited information is available regarding the regulation of motility by glutamic acid at a low external osmotic pressure, it was reported that glutamic acid accumulates rapidly in *E. coli* cells in response to high external osmotic pressure and induces osmoregulatory responses, by which cells prevent or reverse cytoplasmic dehydration[14,15]. Therefore, glutamic acid may also be important for *E. albertii* cells to resist high extracellular osmotic pressure.

In order to further investigate the influence of osmotic stress on *E. albertii* motility, we prepared the M9E culture medium with low salt and a range of osmotic pressures adjusted by the addition of xylose ($0.05$–$0.25 \, \text{osm} \, l^{-1}$). In both HIPH08472 and HIPH14005 strains, the proportions of swimming cells were less than 1% at $0.25 \, \text{osm} \, l^{-1}$ (Fig. 3c), an osmotic pressure close to plasma osmolarity in the humans ($0.29 \, \text{osm} \, l^{-1}$)[16]. At the lower osmotic pressure tested ($0.05 \, \text{osm} \, l^{-1}$), however, the proportion of swimming cells in HIPH14005 cultures reached an average of 48% (Fig. 3c). These results suggest that *E. albertii* cells sense low osmotic pressures and react to this stimulus by inducing flagella formation and motility.

## Conclusion

*E. albertii* was thought to be a non-motile microorganism, which does not produce flagella. To our knowledge, this is the first report showing that flagellar biosynthesis is induced by hypoosmotic stress in enteric bacteria. We demonstrated that 46% of *E. albertii* strains were motile when they were exposed to low osmotic pressure at 20 °C, but not at 37 °C. This induced motility was associated with the induction of flagellar synthesis, which is likely regulated at the transcription level. In *E. coli*, the histone-like nucleoid-structuring (H-NS) protein regulates flagella-related gene expression[17]. The H-NS protein negatively controls expression of many genes regulated by environmental parameters, such as low temperature[18], hence, this protein may regulate the transcription of flagella-related genes in *E. albertii*. Human body temperature (37 °C) suppressed the umbrella-like growth of motile *E. albertii* strains. Their flagella and motility almost disappeared at osmotic pressures as high as that of human plasma ($0.29 \, \text{osm} \, l^{-1}$)[16]. Thus, flagella production would be repressed in the human intestine, which can assist *E. albertii* cells in invading to intestinal epithelial cells. The conditions found to induce *E. albertii* motility (hypoosmotic medium; lower temperatures) often occur in external, aquatic environments. In such environments, although energetically expensive, swimming at the water surface would increase the opportunity of *E. albertii* cells to encounter host animals. Furthermore, flagellated *E. albertii* cells have survival advantages under such conditions over non-flagellated cells. However, it is not clear whether *E. albertii* can perform directional movements to their potential hosts since chemotaxis genes have not been identified to date in this species. Moreover, free-swimming *E. albertii* cells changed direction via tumbling. Tumbling occurs when at least one of the flagellar motors decelerates[19] or reverses direction[20]. In *E. coli* and most other motile bacteria, chemotaxis response regulator CheY and some components of chemotactic signaling are required to induce inverse rotations in the flagellar motor(s), thus promoting transient tumbling to reorient the cell body[21]. Although chemotaxis-related genes remain unknown in *E. albertii*, other factors may regulate flagellar motors in this species.

## Methods

***E. albertii* isolates**. Bacterial strains used and sequenced in this study are listed in Supplementary Table 1. Twenty-nine *E. albertii* strains were described previously[3,22].

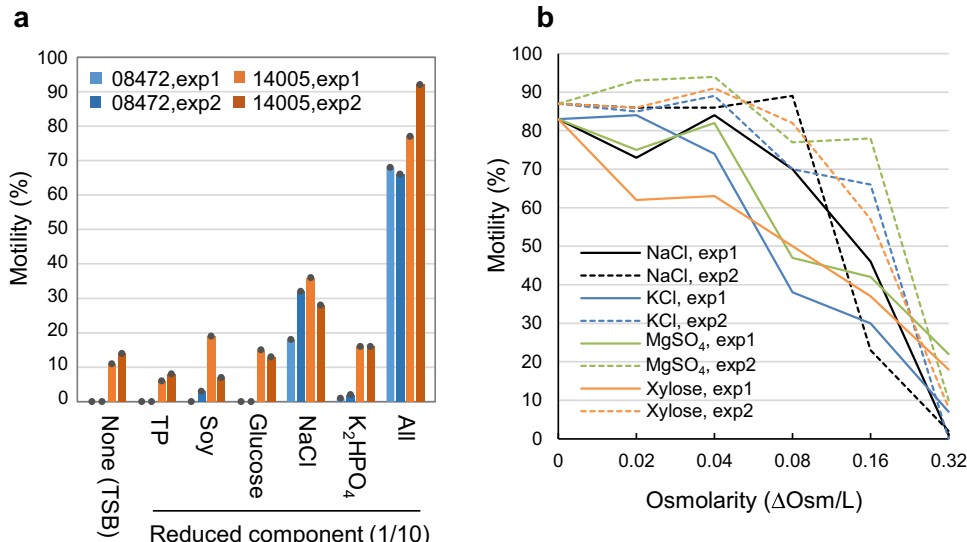

**Fig. 2 Induction of swimming motility in *E. albertii* by low osmotic pressure. a** Induction of motility by reducing NaCl concentration. HIPH08472 and HIPH14005 were cultured for 24 h in ordinary TSB or in several variants of this culture medium containing reduced concentrations of each component. TP trypticase peptone, Soy soybean meal. Results from two independent experiments are shown. **b** Inhibition of motility by high osmotic pressure. HIPH14005 was cultured in media with different osmolarities, achieved by the addition of NaCl, KCl, MgSO₄, or xylose to 1/10TSB. The proportions of swimming cells decreased as osmotic pressure increased, irrespective of the substance added. Results from two independent experiments are shown.

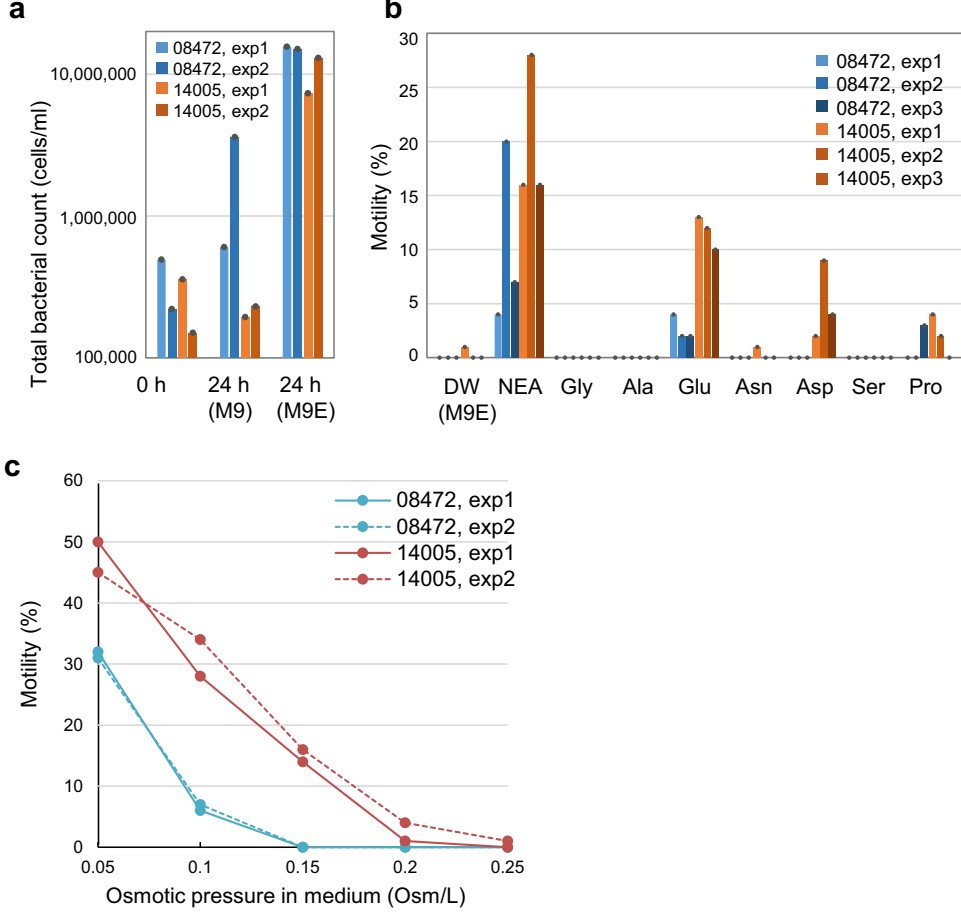

**Fig. 3 Induction of motility in *E. albertii* grown in a chemically defined medium. a** Bacterial cell numbers after 24 h of incubation in M9 (0.24 osm l⁻¹) and M9 supplemented with essential amino acids (M9E) at 20 °C. Results from two independent experiments are shown. **b** Effect of non-essential amino acids (NEA) on *E. albertii* motility. HIPH08472 and HIPH14005 were cultivated in M9E medium supplemented with different non-essential amino acids. The addition of glutamic acid induced motility in both HIPH08472 and HIPH14005 strains. Results from three independent experiments are shown. **c** Induction of motility in *E. albertii* by low osmotic pressure. HIPH08472 and HIPH14005 strains were exposed for 24 h to M9E low-salt medium supplemented with xylose so that a range of osmotic pressures was achieved. Results from two independent experiments are shown.

**Bacterial culture media preparation.** TSB (BD) was used as the basal liquid medium for the maintenance of *E. albertii*. Modified TSBs were prepared by decreasing each of the five TSB components to one-tenth of the original amounts [$17\,g\,l^{-1}$ BBL trypticase peptone, $3\,g\,l^{-1}$ BBL phytone peptone (soybean meal), 2.5 $g\,l^{-1}$ glucose, $5\,g\,l^{-1}$ NaCl, $2.5\,g\,l^{-1}$ $K_2HPO_4$]. For semi-solid agar motility assay, 1/10TBS agar containing a 1/10 amount of TSB and $0.3\,g\,l^{-1}$ agar was used. The M9 minimal medium contained $6\,g\,l^{-1}$ $Na_2HPO_4$, $3\,g\,l^{-1}$ $KH_2PO_4$, $0.5\,g\,l^{-1}$ NaCl, $1\,g\,l^{-1}$ $NH_4Cl$, 0.2% glucose, 1 mM $MgCl_2$, and 0.1 mM $CaCl_2$. To examine amino acid requirements, the M9 minimal medium was supplemented with 1× MEM essential amino acids (Fujifilm wako) (M9E) and/or 1× MEM non-essential amino acids (Fujifilm wako) as indicated. In some cases, glycine, L-alanine, L-asparagine, L-aspartic acid, L-glutamic acid, L-proline, or L-serine was added at $10\,mg\,l^{-1}$ to the M9E medium. The M9E low-salt medium consisted of $750\,mg\,l^{-1}$ $Na_2HPO_4$, 375 $mg\,l^{-1}$ $KH_2PO_4$, $62.5\,mg\,l^{-1}$ NaCl, $125\,mg\,l^{-1}$ $NH_4Cl$, 0.2% glucose, 1 mM $MgCl_2$, 0.1 mM $CaCl_2$, and 1× MEM essential amino acids (Fujifilm wako).

**Swimming motility assay.** Semi-solid agar motility assay was performed by stabbing a 3 ml of semi-solid agar with *E. albertii* culture. The cells were incubated for 72 h at 20 °C or 37 °C. Motility of the cells in the liquid media were observed under the differential interference contrast microscopy after culturing cells in 3 ml of 1/10TSB in test tube for 18–24 h at 20 °C. To assess the proportion of swimming cells, the cells were cultured in 500 μl of media in a 24-well plate for 18–24 h at 20 °C. Movies were taken for 20 s and more than 100 cells that were not attached on the slide glass were observed. The cells showing obvious active movement were judged as motile.

**Bacterial adhesion and invasion assay.** Evaluation of bacterial association and invasion was performed as described by Pacheco et al.[23] with modifications. Briefly, for induction of flagella, 10 μl of *E. albertii* grown in TSB at 37 °C overnight were cultured in a 10 cm dish containing 18 ml of 1/10TSB at 20 °C for 18 h. To prepare non-flagellated bacteria, the bacteria were cultured in a 50 ml tube containing 18 ml of TSB at 37 °C for 18 h. Caco-2 cells were grown for 4 days, washed three times with phosphate-buffered saline (PBS), and infected with ~$10^7$ colony-forming units (CFU) of bacteria in culture medium for 2 h. To assess the total number of cell-associated bacteria, the infected cells were washed three times with PBS and lysed in 1% Triton X-100 in buffered peptone water consisted of $10.0\,g\,l^{-1}$ of peptone, $5.0\,g\,l^{-1}$ of NaCl, $3.5\,g\,l^{-1}$ of $Na_2HPO_4$, and $1.5\,g\,l^{-1}$ of $KH_2PO_4$. The bacteria in the cell lysate was diluted with buffered peptone water and quantified by plating serial dilutions onto 3M Petrifilm E. coli/Coliform Count Plates (3M). To obtain the number of intracellular bacteria, a second set of infected cells was incubated with $200\,\mu g\,ml^{-1}$ gentamicin (Sigma) in medium for 1 h at 37 °C. After washing five times with PBS, cells were lysed and the number of bacteria was assessed as described above.

**Scanning electron microscopy.** Scanning electron microscopy (SEM) images were obtained using the JSM 5600LV (JEOL Ltd, Tokyo, Japan) and the FlexSEM1000 (Hitachi Ltd, Tokyo, Japan). *E. albertii* cells from overnight culture were collected by centrifugation ($100 \times g$), washed with saline, and 20 μl of the cell suspension was applied to the APS-coated cover glasses ($6 \times 6$ mm). The cells were fixed by the addition of 5 μl of glutaraldehyde and incubated for 15 min. The cover glasses were dehydrated through a graded series of ethanol and the ethanol was substituted to t-butanol. The cover glasses were attached to adhesive carbon pad on SEM stubs and frozen for about 30 min. After sublimation, samples were subjected to vapor deposition of gold and observed by SEM.

**RNA extraction and qRT-PCR.** Total RNA was extracted using the RNeasy Mini Kit (Qiagen) according to the manufacturer's instruction. The expression levels of mRNAs were examined by qRT-PCR using the RNA-direct SYBR green real-time PCR master mix (Toyobo). Primers used in this work are listed in Supplementary Table 2.

**Genome sequencing of E. albertii strains.** Genomic DNA was purified from a 2-ml overnight culture using a DNeasy Blood and Tissue kit (Qiagen) according to the manufacturer's instructions. Sequencing libraries for each strain were prepared using the Nextera XT DNA Sample Prep kit (Illumina) to obtain pair-end sequences ($300\,bp \times 2$) on the Illumina MiSeq platform. Draft genome sequences were obtained by assembling the read sequences using Platanus version 1.1.414 (ref. [24]).

**Analyses of flagellar biosynthesis-related genes.** Four of flagellar biosynthesis-related regions of five *E. albertii* strains sequenced in this study were identified from draft genomes by blastn search using the *flhA*, *flgG*, *fliI*, and *fliD* sequences as queries. Protein coding sequences (CDSs) and their functions were predicted using the in silico Molecular Cloning Genomics Edition software (IMC-GE; In Silico Biology, Inc.).

**Statistics and reproducibility.** Statistical analysis of two data sets was performed using two-tailed unpaired Student's *t*-test. A *p* value of 0.05 was considered statistically significant. All experiments were repeated two or three times (see figure legends).

**Reporting summary.** Further information on research design is available in the Nature Research Reporting Summary linked to this article.

## Data availability
The genome assembly data of four *E. albertii* strains obtained in this study are available through GenBank/EMBL/DDBJ under the BioProject accession number PRJDB8602. The assembles and read sequences have been deposited under the accession number BLJJ01000001–BLJJ01000253, BLJK01000001–BLJK01000231, BLJL01000001–BLJL01000353, and BLJM01000001–BLJM01000228. Other data supporting the findings of this study are available within the paper and its supplementary information files.

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

## Acknowledgements
We thank Masahiro Miyoshi for help with Caco-2 cell culture and Katsumi Tamada and Noriko Yamada for collecting fecal droppings of wild birds.

## Author contributions
T. Ikeda conceived of the project. T. Ikeda and T.S. designed experiments. T. Ikeda conducted all experiments with supports of T. Ito (electron microscopic examinations), T.O, J.N., Y.G., Y.O., and T.H. (genome sequencing), Y.O. and A.K. (other experiments). T. Ikeda. and T.O. provided bacterial strains. T. Ikeda and T.S. wrote the manuscript with contributions from T.H. All authors reviewed the manuscript.

## Competing interests
The authors declare no competing interests.
