## [Peer Review File · Communications Biology]

Reviewers' comments:

Reviewer #1 (Remarks to the Author):

This is an interesting article presenting new findings on the swimming behavior of *Escherichia albertii*. It is well written and figures are of high quality, and I concur with authors conclusions and speculation regarding the reason for *E.alb* swimming behavior.

Suggestions for minor revisions:

In abstract "augmented" is too vague – I would recommend changing it to indicate more clearly the direction of the change in the proportion of swimming cells

Lines 90-91: the typical run-and-tumble motion to the best of my understanding has evolved to allow chemotactic movement. This indicates that *E.alb* might actually be able to sense the environment and use that to guide the direction of its swimming. It might be valuable to discuss this in the discussion section.

Reviewer #2 (Remarks to the Author):

The manuscript is well written, and data presented are new and of great interest to the community as they improve the knowledge on the possible modes of dispersion of the emerging enteropathogen *E. albertii*. The conclusions are clear and supported by the results obtained.

Reviewer #3 (Remarks to the Author):

In the presented study, Ikeda et al., performed a preliminary analysis of *Escherichia albertii* in terms of motility. They found that 27 out of 59 *E. albertii* strains are motile when cultured under certain conditions, contrasting the non-motile characteristics reported before on one isolate. The content of this paper is description of this phenomenon, which the authors consider as the novel and significant. However, this reviewer holds a different opinion. The flagellar systems in bacteria, especially in *Escherichia*, are highly conserved and among the gene clusters that can be identified without any difficult. As mentioned in the introduction, 74% of *E. albertii* strains possess a complete set of flagellar biosynthesis genes. Thus, the findings that some of strains have motility are expected. More critically, the work does not contain much physiological characterization, or whatsoever, and efforts to link the observed motility with differences in genomes of the test strains, with respect to flagellar systems, were not seen. In short, they failed to present insights into their observation, which may have a true scientific value.

Reviewer #4 (Remarks to the Author):

General comments:

In view of the important roles of motility in the environmental adaptation and pathogenicity of bacteria, the expression regulation of flagellar genes and their relationship with pathogenicity have been well studied. The manuscript titled "Hypoosmotic stress induces flagellar biosynthesis and swimming motility in *Escherichia albertii*" by Tetsuya Ikeda et al. reported the regulation of flagellar

biosynthesis and motility in *E. albertii* by their internal and external osmolarity. The manuscript presents many interesting and valuable results and the science underpinning it is convincing. However, the manuscript needs to be revised and improved before it can be published. Specific comments:

1. It is generally believed that motility contributes greatly to the pathogenicity of bacteria. The authors are encouraged to carry out some experiments to see if the expression of flagella genes of bacteria is related to pathogenicity, such as adhesion and survival in macrophage. The following two articles are recommended to the authors. (Flagellar motility contributes to the invasion and survival of *Aeromonas hydrophila* in *Anguilla japonica* macrophages. *Fish & Shellfish Immunology*, 2014, 39:273-279; Flagellar motility is necessary for *Aeromonas hydrophila* adhesion. *Microbial Pathogenesis*. 2016.98:160-166.)

2. The expression of flagellar assembly pathway has been reported to be regulated by environmental factors (Involvement of the flagellar assembly pathway in *Vibrio alginolyticus* adhesion under environmental stresses. *Frontiers in Cellular and Infection Microbiology*, 2015, 5, article 59: 1-9). This should be included in the introduction of the manuscript.

3. The number of literature cited by the author is small, and the literature cited is slightly out of date, which can not reflect the latest research progress.

We thank the reviewers for their helpful remarks and thorough review of the manuscript. Below we address each of the comments/concerns raised.

Reviewer #1 (Remarks to the Author):

This is an interesting article presenting new findings on the swimming behavior of Escherichia albertii. It is well written and figures are of high quality, and I concur with authors conclusions and speculation regarding the reason for E.alb swimming behavior.

Suggestions for minor revisions:

In abstract “augmented” is too vague - I would recommend changing it to indicate more clearly the direction of the change in the proportion of swimming cells

Thank you, we have added the following sentence to the abstract (line 17).
“..., the addition of glutamic acid, an amino acid known to regulate the internal cell osmolarity, augmented the proportion of swimming cells by 35-fold.”

Lines 90-91: the typical run-and-tumble motion to the best of my understanding has evolved to allow chemotactic movement. This indicates that E.alb might actually be able to sense the environment and use that to guide the direction of its swimming. It might be valuable to discuss this in the discussion section.

We thank the reviewer for this suggestion and have now added following discussion to the text (lines 225-232).
“Moreover, free-swimming *E. albertii* cells changed direction via tumbling. Tumbling occurs when at least one of the flagellar motors decelerates¹⁹ or reverses direction²⁰. In *E. coli* and most other motile bacteria, chemotaxis response regulator CheY and some components of chemotactic signaling are required to induce inverse rotations in the flagellar motor(s), thus promoting transient tumbling to reorient the cell body²¹. Although chemotaxis-related genes remain unknown in *E. albertii*, other factors may regulate flagellar motors in this species.”

Reviewer #2 (Remarks to the Author):

*The manuscript is well written, and data presented are new and of great interest to the community as they improve the knowledge on the possible modes of dispersion of the emerging enteropathogen *E. albertii*. The conclusions are clear and supported by the results obtained.*

We thank the reviewer for their positive remarks.

Reviewer #3 (Remarks to the Author):

*In the presented study, Ikeda et al., performed a preliminary analysis of *Escherichia albertii* in terms of motility. They found that 27 out of 59 *E. albertii* strains are motile when cultured under certain conditions, contrasting the non-motile characteristics reported before on one isolate. The content of this paper is description of this phenomenon, which the authors consider as the novel and significant. However, this reviewer holds a different opinion. The flagellar systems in bacteria, especially in *Escherichia*, are highly conserved and among the gene clusters that can be identified without any difficulty. As mentioned in the introduction, 74% of *E. albertii* strains possess a complete set of flagellar biosynthesis genes. Thus, the findings that some of strains have motility are expected.*

We agree with the reviewer that it has been presumed that a part of *E. albertii* strains are motile in certain conditions. However, there has been no report of success in finding motile *E. albertii* cells before. One reason would be an unprecedented feature of flagellar regulation in *E. albertii*. We found that hypoosmotic stress is required for its flagellar induction. In *E. albertii*, flagellar biosynthesis is repressed at osmolarity as high as ordinary medium such as TSB. The addition of nutrients or other compounds inhibits their synthesis further and makes it difficult to induce flagella.

More critically, the work does not contain much physiological characterization,

or whatsoever, and efforts to link the observed motility with differences in genomes of the test strains, with respect to flagellar systems, were not seen. In short, they failed to present insights into their observation, which may have a true scientific value.

We agree with the reviewer that physiological characterization should be included. Following the advice of reviewer #4, we performed bacterial adhesion and invasion assay. This revealed that induction of flagella increased survival of *E. albertii* cells in the intestinal epithelial cell culture in the presence of gentamicin. (lines 111-125, Table 2).

In lines 99-109, we performed the whole genome sequencing and confirmed that among six strains sequenced two non-motile strains had mutations in flagellar-related genes but the others did not.

Reviewer #4 (Remarks to the Author):

General comments:

In view of the important roles of motility in the environmental adaptation and pathogenicity of bacteria, the expression regulation of flagellar genes and their relationship with pathogenicity have been well studied. The manuscript titled "Hypoosmotic stress induces flagellar biosynthesis and swimming motility in Escherichia albertii" by Tetsuya Ikeda et al. reported the regulation of flagellar biosynthesis and motility in E. albertii by their internal and external osmolarity. The manuscript presents many interesting and valuable results and the science underpinning it is convincing. However, the manuscript needs to be revised and improved before it can be published.

Specific comments:

1. It is generally believed that motility contributes greatly to the pathogenicity of bacteria. The authors are encouraged to carry out some experiments to see if the expression of flagella genes of bacteria is related to pathogenicity, such as adhesion and survival in macrophage. The following two articles are recommend to the authors. (Flagellar motility contributes

to the invasion and survival of Aeromonas hydrophila in Anguilla japonica macrophages. Fish & Shellfish Immunology, 2014, 39:273-279; Flagellar motility is necessary for Aeromonas hydrophila adhesion. Microbial Pathogenesis. 2016.98:160-166.)

We thank the reviewer for this suggestion. We performed bacterial adhesion and invasion assay using human intestinal cell line Caco-2. This revealed that induction of flagella increased survival of *E. albertii* cells 130-fold in the cell culture in the presence of gentamicin. The results have been added to the text (lines 111-125, Table 2).

2. The expression of flagellar assembly pathway has been reported to be regulated by environmental factors (Involvement of the flagellar assembly pathway in Vibrio alginolyticus adhesion under environmental stresses. Frontiers in Cellular and Infection Microbiology, 2015, 5, article 59: 1-9). This should be included in the introduction of the manuscript.

Thank you, the paper has been included in the introduction of the manuscript (lines 41-43).

3. The number of literature cited by the author is small, and the literature cited is slightly out of date, which can not reflect the latest research progress.

Thank you, recent information about flagellar ejection and rotation has been added to the manuscript (lines 39-41, 225-232).

REVIEWERS' COMMENTS:

Reviewer #3 (Remarks to the Author):

The revision did not provide physiological characterization that is sufficiently strong.

Reviewer #4 (Remarks to the Author):

After revision, the quality of the manuscript has been significantly improved. I think the manuscript can be accepted for publication.